# Measurement invariance tests of revisions to archaically worded items in the Mach IV scale

**Brian K. Miller** *, Kay Nicols, Robert Konopaske

Department of Management, Texas State University, San Marcos, Texas, United States of America

* bkmiller@txstate.edu

**Data Availability Statement:** The online Texas Data Repository (dataverse.tdl.org/dataverse/txstate) is used to share datasets through the Texas Digital Library and managed by local Texas State University librarians. The Texas Digital Library (TDL) is a consortium of academic libraries in

## Abstract

The Machiavellian IV [1] instrument, developed almost 50 years ago to measure trait Machiavellianism and still in wide use in personality research, uses item wording that is not gender-neutral, makes use of idiomatic expressions, and includes archaic references. In this two-sample study, exploratory factor analysis (EFA) was conducted on one sample to examine the structure of responses to the Mach IV. In an independent second sample the resulting EFA structure was analyzed using confirmatory factor analysis-based measurement equivalence/invariance (ME/I) tests in a control group with the original archaic items and a treatment group with eight items rewritten in a more modern vernacular. Specific model testing steps [2] and statistical tests [3] were applied in a bottom-up approach [4] to ME/I tests on these two versions of the Mach IV. The two versions were found to have equal form, equal factor loadings, but unequal indicator error variances. Subsequent item-by-item tests of error invariance resulted in substantial decrements to fit for three revised items suggesting that the error associated with these items was not equal across the two versions.

## Introduction

Machiavellianism is the predisposition to manipulate interpersonal relationships with guile, opportunism, and deceit. Research on this construct began in earnest with the development of a self-report inventory [1] based upon the lead character in Niccolo Machiavelli's sixteenth century novel "The Prince". The title character lacked morality and empathy and had ample distrust of others. The scale authors [1] painstakingly developed items designed to measure the lead character's tendency toward amorality, to have negative views of human nature, and to employ interpersonal tactics designed for personal gain at the expense of others [1]. The result was the Mach IV which is a 20-item scale comprised of three subscales: morals, views, and tactics. This instrument is the most frequently used scale to measure Machiavellianism [1].

Interest in Machiavellianism was further with its inclusion as one of the three traits comprising the relatively new Dark Triad [5]. The other two traits are narcissism and psychopathy. Machiavellianism is distinct from the others in its focus on the strategic manipulation of others for personal gain [6]. Machiavellianism is the most often studied but least understood of the Dark Triad traits [6] likely due to its psychometric complexity resulting from factor indeterminacy, poorly loading items, and subscale unreliability. Numerous attempts at exploratory factor analysis (EFA) have revealed a factor structure ranging from the original three factors [1]

Texas with a proven history of providing shared technology services to support secure, reliable access to digital collections of research and scholarship. The Texas Data Repository is a project of the TDL and its member institutions to develop a consortial statewide research data repository for researchers at Texas institutions of higher learning. Data is curated in the repository following accepted standards (NISO Framework Advisory Group, 2007). The persistent identifier, a DOI, used for the data in this study is https://dataverse.tdl.org/dataset.xhtml?persistentId=doi:10.18738/T8/WPZSAP.

**Funding:** The authors received no specific funding for this work.

**Competing interests:** The authors have declared that no competing interests exist.

to as many as nine factors [7], due in part to the diversity of populations to which it has been administered. Confirmatory factor analyses (CFA) have required the elimination of items ranging from seven of 20 [8] to ten of 20 [9] in order to achieve adequate model fit and/or model convergence. These problems with the Mach IV are not unlike other lengthy scales with obliquely related subscales and have greatly contributed to the limited understanding of Machiavellianism.

It is possible that problems with the Mach IV arise, at least partially, from outdated language not understandable to contemporaneous survey responders. The English lexicon is constantly changing as new words are created and outdated ones fall out of favor. Developers of self-report inventories are wise to use the vernacular of the time to make their instruments understandable but not so colloquial or era-specific so that they are only appropriate for contemporaneous respondents. Scales developed decades or generations ago can be particularly problematic for at least three reasons. First, because samples of college students are commonly used in psychological research [10,11], it is imperative that the wording of scale items is understandable to young adults and terms that refer to popular culture, historical events, or figures of the distant past are likely to be confusing or un-interpretable by many of today's college students. Second, from 1970–2017 the percentage of female students enrolled in college escalated from 48.5% to 56.4% of all students [12]. With this in mind, items referring only to men or that have prototypically masculine inferences are perhaps not conceptualized the same by females as by male respondents and are problematic given that the majority of research participants are likely to be female. Third, concurrent with changes in gender demography amongst college students is an increase in the number of international students enrolled in American universities which has quadrupled since 1976 (the earliest year data were tracked), topping one million in 2016 [13]. The largest numbers of international students are from China (31.5%), India (15.9%), Saudi Arabia (5.9%), and South Korea (5.8%) [14,15]. Some of these non-native English speaking college students are also likely participating in psychological research. For non-native English speakers, the inclusion of idiomatic expressions whose literal meaning (e.g., s/he is taking a major risk) is not easily discernable from the figurative meaning (e.g., s/he is playing with fire) can lead to misinterpretations or misunderstandings [16,17]. The psychometric properties of scales developed decades ago may therefore suffer lower validity and reliability when administered to current samples.

The current study examines whether a revised Mach IV scale [1] that omits archaic language and outdated references, eliminates idiomatic or unusual expressions, and corrects for non-gender-neutral language will be conceptualized differently than the original version developed almost 50 years ago. To this end, the factor structure of the original Mach IV scale was examined with exploratory factor analysis (EFA) in one sample and the structure was validated with confirmatory factor analysis (CFA) in an independent second sample. In the second sample CFA-based measurement equivalence / invariance (ME/I) tests were used to compare the original and revised items in order to determine if respondents in the two different groups ascribe the same meaning to items in both versions of the scale [2]. The ME/I tests are used to determine if different versions of a scale measure the same construct, in the same manner, in different groups [18]. It is hoped that the psychometric properties of the revised scale are better than those of the original scale developed decades ago.

There does exist an alternative version of the Mach IV known as the Kiddie Mach that was also developed [1] for administration to children. To address that audience, the authors discarded the original item on euthanasia and completely changed the intent and meaning of other items for such a young audience but some problematic item wording remained. The adult Mach IV instrument has enjoyed much more widespread use, however, and is thus the focus of this study.

**Table 1. Previous exploratory factor analytic results for the Machiavellianism IV scale [1].**

| Item # | Christie & Geis (1970) | Williams, et al. (1975) | Kuo & Marsella (1977) Chinese sample | Kuo & Marsella (1977) US sample | Ahmed & Stewart (1981) | O'Hair & Cody (1987) | Panitz (1989) | Andreou (2004) |
|---|---|---|---|---|---|---|---|---|
| | | | | Factor on which item loaded | | | | |
| 1. | 1 | 2 | 4 | 2 | 1 | 2 | 6 | 3 |
| 2. | 1 | 2 | 5 | 5 | 3 | 1 | 4 | 3 |
| 3. | 1 | 4 | 5 | 4 | 1 | 2 | 1 | 4 |
| 4. | 2 | 3 | 1 | 3 | 1 | -- | 2 | 2 |
| 5. | 2 | -- | 1 | 1,2,5 | 3 | 1 | 3 | 3 |
| 6. | 1 | 1 | 1 | 1,3 | 5 | 2 | 2 | 4 |
| 7. | 1 | 1 | 3 | 2 | 1,5 | 2 | 1 | -- |
| 8. | 2 | 2 | 2 | 1 | 4 | -- | 2 | 3 |
| 9. | 3 | 1 | 4 | 2 | 1 | 2 | 5 | 1 |
| 10. | 1 | 1 | 2 | 1,2 | 3 | 1 | 3 | 3 |
| 11. | 2 | -- | -- | 1 | 5 | -- | 3 | 2 |
| 12. | 1 | -- | 2 | 5 | 4 | -- | 7 | -- |
| 13. | 2 | -- | 5 | 3 | -- | -- | 6 | 1 |
| 14. | 2 | 3 | 2, 3 | 3 | 2 | 3 | 2 | 1 |
| 15. | 1 | 2 | 2 | 5 | 3,4 | 1 | 7 | -- |
| 16. | 1 | -- | 3 | 4 | 5 | 2 | 5 | 2 |
| 17. | 2 | -- | 2 | 3,4 | 2,4 | 3 | 1 | 1 |
| 18. | 2 | -- | -- | 4 | 5 | 2 | 1 | 2 |
| 19. | 3 | 4 | 3 | 3,4 | -- | 2 | 1 | 4 |
| 20. | 2 | 2 | 2 | 1,2 | 2 | 3 | 4 | 1 |
| Type of EFA: | | PCA | PAF | PAF | PCA | PCA | Unknown | PCA |

Note: PCA = principal components analysis; PAF = principal axis factoring; Cells with more than one number indicate cross-loadings. Double dash indicates the item failed to load on any factor.

Over the decades, various EFA examinations of the adult Mach IV have been undertaken. The factor structure of the Mach IV has ranged from the original three-factor structure [1] to a nine-factor structure [7]. These disparate solutions have also resulted in items in one solution loading on another factor in another solution. For example, item nine of the Mach IV ("It is wise to flatter important people") sometimes loads on the third factor [1]; on the first factor [19,20,21] on the second factor in U.S. samples [22,23], on a fourth factor in a Chinese sample [22], and even on a fifth factor [7]. This variety of factor solutions and very different loadings of items on factors in the Mach IV instrument suggests that further factor analyses are in order to build an understanding of the underlying factor structure of the Mach IV. See Table 1 for these factor solutions.

In partial response to these varying solutions to the Mach IV instrument, alternative measures of Machiavellianism have been developed. For example, some researchers have developed a multi-factor measure of Machiavellianism known as the Machiavellianism Personality Scale [24]. It too is not without problems [25,26,27]. Other researchers have developed promising new conceptualizations and measures of Machiavellianism [28,29]. Rather than developing another new measure of Machiavellianism, the current study seeks to update the Mach IV instrument by rewriting some problematic items and examining the factor structure of a revised version of the Mach IV in comparison to the structure of the original version.

## Study one method

### Procedure

Studies One and Two were approved by the Institutional Review Board (IRB) of Texas State University with documentation from the IRB Regulatory Manager. The approval number was EXP2016H133885Y and it was declared to be exempt from review. Consent from participants was gained verbally and in writing with signatures on forms that were separate from the actual surveys. Thus, all data were collected anonymously.

In Study One, data were collected via an anonymous paper-and-pencil self-report inventory in large sections of an upper level undergraduate course at a large public university in the southwestern U.S. Because of the many different factor structures found in previous research for both instruments, the purpose of Study One was to use EFA to determine the latent structure of the Mach IV instrument [30]. The resulting factor structure is validated with CFA in Study Two. Given the many issues that can affect EFA results (e.g. communalities, rotation method, sample size), the decision of an appropriate cutoff score for factor loadings is vital to interpreting the factor structure. Despite popular arguments against the use of strict statistical cutoffs [31,32] it is still widely accepted [33,34,35,36] that factor loadings should be at least .30 to be minimally acceptable [30]. This cutoff was used in this study in a principal axis factoring analysis on the data collected in Study One to extract a factor solution for the Mach IV instrument with an oblique rotation because the sub-scales of the instrument should be correlated.

### Participants

Complete data were provided for Study One by 295 respondents. Listwise deletion was used so four participants who failed to complete the survey were discarded from the analysis. Slightly more than half of the participants were female (52.0%). The mean age was 21.97 years and self-reported racial or ethnic group membership was 59.7% White, 6.8% Black, 27.8% Hispanic, 2.4% Asian, 0.7% American Indian, and 2.7% other. The mean level of full-time and part-time work experience was 18.19 months and 41.98 months, respectively. Of the nearly 63% of respondents (i.e. 185) who were currently employed, 21% were direct supervisors or managers of other employees. Of those 33 supervisors or managers, the mean number of direct reports was nine with a range of two to 40.

### Mach IV instrument

Responses were gathered on the 20-item Mach IV scale [1] using a Likert response scale anchored by 1 = "strongly disagree" and 7 = "strongly agree". Sample items included: "It is hard to get ahead without cutting corners here and there" and "Barnum was very wrong when he said there's a sucker born every minute" (reverse scored). All items were corrected for reverse scoring before the EFA was conducted. Cronbach's coefficient alpha of internal consistency reliability for scores on the Mach IV in Study One was .68. The item-level skewness and kurtosis ranged from -1.12 to 1.87 and from -1.08 to 4.04, respectively. These univariate statistics were within acceptable limits of |2.0| for skewness and |7.0| for kurtosis, respectively [37].

### Study one results

A direct oblimin rotation with principal axis factor extraction resulted in a six-factor solution for the Mach IV that explained 50.21% of the variance in the items. The number of factors was based upon a visual inspection of the scree plot, the criteria of Eigen values greater than one, and parallel analysis. Sixteen of 20 items had factor loadings greater than .30 on one factor only (i.e. clean loadings), two items had cross loadings of greater than .30 on two factors, and

**Table 2. Factor loadings for the Mach IV with principal axis factor analysis and an oblique rotation.**

| Items | Factors | | | | | |
|---|---|---|---|---|---|---|
| | 1 | 2 | 3 | 4 | 5 | 6 |
| Mach1 | -.003 | .283 | .065 | .141 | -.148 | .001 |
| Mach2 | **.314** | -.043 | .193 | .000 | -.378 | -.288 |
| Mach3 | -.032 | .020 | -.09 | -.041 | .070 | **-.566** |
| Mach4 | .080 | **.495** | -.206 | -.004 | -.009 | .070 |
| Mach5 | **.676** | .025 | -.099 | .007 | -.016 | .002 |
| Mach6 | -.044 | **.407** | -.074 | -.211 | -.143 | -.228 |
| Mach7 | .119 | **.401** | .146 | -.065 | .109 | -.032 |
| Mach8 | **.460** | -.017 | .044 | -.071 | .010 | .032 |
| Mach9 | -.004 | .101 | .100 | **.428** | .124 | -.072 |
| Mach10 | **.318** | -.176 | -.175 | .186 | .012 | **-.425** |
| Mach11 | .126 | .131 | -.080 | .158 | .042 | -.176 |
| Mach12 | .085 | -.008 | .088 | .020 | **.381** | -.075 |
| Mach13 | **.455** | .084 | -.120 | .008 | .072 | .021 |
| Mach14 | -.015 | .056 | **-.605** | -.129 | .048 | -.082 |
| Mach15 | **.468** | .075 | -.132 | .170 | .024 | -.061 |
| Mach16 | .042 | .105 | -.010 | **-.560** | .125 | -.147 |
| Mach17 | .132 | -.076 | **-.497** | -.043 | -.064 | .047 |
| Mach18 | -.060 | .230 | .136 | -.147 | -.006 | **-.316** |
| Mach19 | -.046 | **.410** | -.082 | .081 | .137 | -.105 |
| Mach20 | .093 | .076 | **-.387** | .052 | -.062 | -.087 |

*Note*: Strongest factor loadings are underlined. Factor loadings > .30 in bold and underlined

two items loaded on no factor (i.e. no loadings on any factor > .30). Factors three and six were comprised solely of negatively loading items and factor four had one negatively and one positively loading item in excess of the cutoff. See Table 2 for the factor loadings.

## Study one discussion

The results of Study One suggest that there are some problems with responses to the Mach IV. First, alpha serves as the lower bound for score reliability and it is usually higher for scores on items in a unidimensional scale than in a scale with several sub-scales [38] but the low alpha of .68 here is perhaps symptomatic of the difficulty of estimating a baseline true score resulting from scale multidimensionality. These sub-scales may measure only moderately related dimensions. If the items themselves are not unidimensional then cross-loading items in a multi-factor solution is likely a source of model misfit. On the other hand, if the issue is one of simply misunderstanding the content of the items by respondents then the multidimensionality of the scale is much less of a problem. Second, the number of negatively loading items is potentially problematic. Negatively loading items that measure the opposite end of the Machiavellianism spectrum detrimentally affect overall scale scores. The result is that the sum of the potentially offsetting items can mask important individual differences measured by the scale. That is, items with negative loadings tend to measure the opposite end of the Machiavellianism spectrum and when summed with positively loading items the overall scale score is closer to the middle range than either the low or high ranges. Therefore, the Mach IV is unlikely to be effective in discriminating low or high Machs from mid-range Machs. Whether these negative

**Table 3. Original problematically worded scale items and revised versions examined in study two.**

| Item number | Original Mach IV items | Revised Mach IV items |
|---|---|---|
| 2 | When you ask someone to do something for you, it is best to give the real reasons for wanting it rather than giving reasons which might <u>carry more weight</u> | When you ask someone to do something for you, it is best to give the real reasons for wanting it rather than giving reasons which might <u>be more influential</u> |
| 4 | It is hard to get ahead without <u>cutting corners</u> here and there | It is hard to get ahead without <u>taking the easy way</u> here and there |
| 6 | It is safest to assume that all people <u>have a vicious streak and it will come out</u> when they are given a chance | It is safest to assume that all people <u>can be cruel</u> when they are given a chance |
| 9 | It is wise to <u>flatter</u> important people | It is wise to <u>be especially nice</u> to important people |
| 11 | <u>Barnum was very wrong when he said there's a sucker born every minute</u> | <u>It is wrong to think that most people can be easily tricked</u> |
| 16 | Most <u>men</u> forget more easily the death of <u>their father</u> than the loss of their property | Most <u>people</u> forget more easily the death of <u>a parent</u> than the loss of their property |
| 18 | Generally speaking, <u>men</u> won't work hard unless they're forced to do so | Generally speaking, <u>people</u> won't work hard unless they're forced to do so |
| 20 | Most <u>men</u> are brave | Most <u>people</u> are brave |

*Note*: Problematic item parts and their revisions are underlined.

factor loadings and low internal consistency reliability are the result of respondents being unsure of the archaic references, uncomfortable with the non-gender-neutral wording, or unfamiliar with the idiomatic/unusual phrasing is not yet known. To gather further evidence in Study Two the CFA tests of the factor structure determined with EFA in Study One were conducted on data from an independent sample administered as an experiment.

## Study two method

### Procedure

In Study Two, the same method of administration at the same university as above was used with different respondents. However, an experimental framework [39,40] was utilized whereby only the participants in the control group received questionnaires in the scale's original language [1]. Participants in the treatment group received surveys with the same instructions as the control group in Study One and most of the same items but with eight of 20 Mach IV items rewritten to avoid problematic language. This study's three authors independently selected specific items from the entire scale, revised those items, and came to a mutual agreement on the number of items to be reworded as well as the exact rewording of the items. For example, the original item "It is hard to get ahead without cutting corners here and there" was revised to "It is hard to get ahead without taking the easy way here and there." Additionally, the original item "Barnum was very wrong when he said there's a sucker born every minute" (reverse scored) was revised to "It is wrong to think that most people can be easily tricked") (reverse scored). All items were corrected for reverse scoring before the CFA was conducted. See Table 3 for a side-by-side comparison of the original and revised items.

In the CFA model tests that follow, the factor structure arising from the previous EFA test was imposed on the factor structure with items loading less than .30 in the EFA being forced to zero in the CFA. The goal was to use as parsimonious of a model of the data as possible. Therefore error terms was not allowed to correlate and items were not allowed to cross-load on multiple factors.

Several CFA-based ME/I tests were implemented in Study Two to determine if the revised items were interpreted differently in the treatment group than the original archaically worded items in the control group. This involved a specific sequence of tests of two broad types: measurement level invariance and latent construct invariance (sometimes referred to as structural invariance). Measurement invariance tests must precede structural invariance tests although methodologists disagree on the specific order of the tests within these two broad categories [41,42,43]. The sequence of steps in the bottom-up approach to ME/I tests [4] were used in the following order: (1) tests of equal form, (2) tests of equal factor loadings, (3) tests of equal indicator error variances, (4) tests of latent factor variance, and (5) tests of the covariance between latent factors comprising the scales. Steps one and two are essential for invariance testing but steps three through five are widely regarded as stringent and not always required [42,2,44]. Evidence of invariance is provided by an examination of changes in model fit from one test to the next.

When researchers examine instruments using ME/I tests it is recommended that they supplement the chi-square difference test, which is known to be heavily influenced by sample size, with other fit indices [45]. For example, a commonly used rule of thumb for changes in model fit for the comparative fit index (CFI) [46] is that each sequentially more restrictive model should result in less than a -0.01 decrement to fit in order to indicate invariance across groups [3]. If the CFI decreases in magnitude in a successively restrictive model by less than -0.01 (e.g., $\Delta$CFI = -.009) then the two models are equivalent. Because there are no standard errors associated with the $\Delta$CFI, this rule of thumb only serves as a guideline and not as a strict statistical test. To establish model fit in the first test of ME/I both the chi-square and the CFI were calculated as baselines. These supplemental tests to the baseline model were the Root Mean Squared Error of Approximation (RMSEA) [47] and the Standardized Root Mean Squared Residual (SRMR). Good model fit is indicated when CFI $\geq$ .95, RMSEA < .06, and SRMR < .08 [48]. More lenient cutoffs are that CFI $\geq$ .90, RMSEA < .10, and SRMR < .10 will indicate good fit [44]. Some use a different cutoff for the RMSEA such that < .08 indicates reasonable fit [47]. With such varying rules of thumb in mind, a strict adherence to fit index cutoffs for the rejection of models should be considered only in light of theoretical or substantive issues [49,50].

## Participants

Complete data were provided anonymously by 483 respondents. Listwise deletion was used so 15 participants who failed to complete the survey were discarded from the analysis. As in Study One, most of the participants were female (52.0%). The mean age was 21.75 years and self-reported racial or ethnic group membership was 60.4% White, 8.5% Black, 25.6% Hispanic, 2.9% Asian, 0.2% American Indian, and 2.3% other. The mean level of full-time and part-time work experience was 18.88 months and 39.26 months, respectively. Nearly 58% were currently employed. Of those 282 currently employed participants, 17% were the direct supervisor or manager of other employees. Of those 48 managers, the mean number of direct reports was 11 with a range of two to 46.

## Mach IV instrument

In the control group ($n$ = 243) with originally worded items, alpha reliability for scores on the Mach IV was .70. The item-level skewness and kurtosis for responses to the original items ranged from -0.84 to 1.86 and from -1.01 to 4.14, respectively. In the treatment group ($n$ = 240) with revised items, alpha was .67. The skewness and kurtosis ranged from -1.01 to 2.06 and from -1.16 to 4.67, respectively. In both groups, item level normality met recommended cutoffs [37].

## Study two results

### Tests of normality

Univariate normality is a necessary, but not sufficient, condition for multivariate normality [51], which was calculated as a function of Mardia's normalized coefficient of kurtosis using a macro [52]. The Mardia's normalized coefficient for Study Two was 38.23 which was larger than the recommended cutoff of |3.0| [53,54]. Because the data were not multivariate normal, the Satorra-Bentler scaled chi-square (S-B $\chi^2$) and robust standard errors adjustment to the maximum likelihood method of estimation was used in CFA with Lisrel 8.8 software [55] and the covariance and asymptotic covariance matrices were used as input.

### CFA tests of the Mach IV

**Preliminary single group analysis.**    In the following CFA we refer to the items of the Mach IV as well as the factors that emerged from the EFA by our original numbering system. In the preliminary analysis with the control group examining the six-factor structure of the 20-item Mach IV found via EFA in Study One, the SRMR and RMSEA were good but the CFI was .894 and therefore just slightly below the recommended cutoff [44]. However, Item 12, which is the only item on Factor 5, was a Heywood case which resulted in a negative error variance for the item and model non-convergence. In the treatment group the fit indices were also good with only the CFI again falling a bit short at .860. In the treatment group, the sole item loading on Factor 5 (Item 12) was also a Heywood case. Among the solutions to a Heywood case is to force the item to load on some other factor or dropping it from the analysis. Item 12 was not a revised item and read as "People suffering from incurable diseases should have the choice of being put painlessly to death" in both groups. To examine Item 12's appropriateness for loading on some other factor, the EFA results from Study One were consulted. The standardized factor loadings for item 12 in the EFA results on factors other than Factor 5 were very weak and ranged from -.075 to .088 so the item was instead dropped from further analysis rather than forcing it to load on another factor on which it would have presumably loaded weakly. With this in mind, a five-factor solution to the Mach IV was re-analyzed in the control and treatment groups.

In subsequent analysis with the control group examining the now-five-factor structure of the now-19-item Mach IV, the fit indices improved with the CFI rising above the cutoff at .904. However, the covariance between Factor 4 and Factor 6 was unidentified indicating that the covariance was unstable and not likely to result in any particular best estimate of the data. The item loadings for both of Factor 4's only two items (Items 9 and 16) were close to zero and negative. In the treatment group the fit indices were similar to those of the last iteration with only the CFI falling short of the cutoff. There were no problems with model convergence in the treatment group but Item 16 also loaded very weakly at only .02 on Factor 4. This item was a revised item. In the control group it read as "Most men forget more easily the death of their father than the loss of their property." In the treatment group the revised item was "Most people forget more easily the death of a parent than the loss of their property." Because of problems with Item 16 in both groups, it was dropped and a now-18-item solution was again re-examined in both groups.

In the five-factor structure of the now-18-item Mach IV, the fit indices for the control group were good and the CFI improved to .903. In the treatment group the fit indices were good but the CFI still lagged a bit below the recommended cutoff. However, Item 9 was still problematic in both groups with a negative error variance in both groups so its standard error could not be calculated. Item 9 was a revised item and read as "It is wise to flatter important

**Table 4. Study two tests of Mach IV scale separately in the control and treatment groups.**

| Model | S-B $\chi^2$ | df | CFI | SRMR | RMSEA (90% CI) |
|---|---|---|---|---|---|
| **Control group** | | | | | |
| 20 items, 6 factors | 248.432 | 155 | .894 | .077 | .050 (.038; .061) |
| 19 items, 5 factors | 225.507 | 142 | .904 | .075 | .049 (.037; .061) |
| 18 items, 5 factors | 203.972 | 125 | .903 | .074 | .051 (.038; .064) |
| 17 items, 4 factors | 180.824 | 113 | .914 | .074 | .050 (.036; .063) |
| **Treatment group** | | | | | |
| 20 items, 6 factors | 259.667 | 155 | .860 | .070 | .053 (.042; .064) |
| 19 items, 5 factors | 247.602 | 142 | .856 | .071 | .056 (.044; .067) |
| 18 items, 5 factors | 266.471 | 125 | .853 | .073 | .060 (.048; .072) |
| 17 items, 4 factors | 216.963 | 113 | .856 | .073 | .062 (.049; .074) |

*Note.* S-B $\chi^2$ = Satorra-Bentler scaled chi-square; df = degrees of freedom; SRMR = standardized root mean square residual; CFI = comparative fit index; RMSEA = root mean square error of approximation; 90% CI = 90% confidence interval for RMSEA

people" in the control group and as "It is wise to be especially nice to important people" in the treatment group. Thus, Item 9 was dropped and the model was yet again re-analyzed in both groups.

In the now-four-factor structure of the now-17-item Mach IV, in the control group the fit indices were good and the CFI rose further to .914. In the treatment group the fit indices were also good and although the CFI was .856, that was an improvement over the previous iteration. There were no problematic items and the model converged in both groups with no Heywood cases. However, because of sequential item elimination these fit statistics are likely biased upward and subsequent analysis based on these shortened scales should be viewed with caution given the changes to the instrument and its factor structure that were required to achieve model fit. Thus, the best fitting CFA solution to the Mach IV instrument used 17 of the original 20 items in a four-factor solution. It is noteworthy that two of the three items discarded were revised items and that this factor structure is not dissimilar to that of previous factor analyses of of the Mach IV instrument which vary wildly from three to seven factors on items ranging in number from 10 to 20. See Table 4 for these fit indices.

**Multiple group (ME/I) analysis.** Before combining the two groups for the ME/I tests, one item for each of the remaining four factors was selected to serve as the referent indicator so as to set the metric of the four latent constructs in the model. The factor variance could not be set to unity (i.e. 1), because setting factor variances to unity in both groups essentially constrains them to equivalency. Tests of factor variance equivalency are the fourth step in the ME/I sequence [4,2] and constraining the factor variances to equality in steps one through three is inappropriate. Therefore, the item that loaded on each factor with the most similar magnitude in both groups was constrained to unity (and therefore to equivalency) in subsequent analyses. These referent items (using the original numbering of the scale) were: Item 15 on Factor 1, Item 1 on Factor 2, Item 14 on Factor 3, and Item 18 on Factor 6 (note that Factors 4 and 5 were eliminated from the model because of problematic items).

The tests of equal form between the two groups resulted in S-B $\chi^2$ = 398.693 (df = 230, p < .001), CFI = .882, and RMSEA = .055 (90% CI: .046, .064). Despite being a bit low, the CFI of .882 served as the baseline model fit statistic to which the second test was compared. The second test of the model was for equal factor loadings (i.e. full metric equivalence) and resulted in an increase in the CFI of .002 which was less than the cutoff recommendation [3]. The third test of equal indicator error variances resulted in $\Delta$CFI = -.010 and therefore did not meet the

**Table 5. Study two tests for measurement invariance for a four factor solution for 17 of the 20 items on the Mach IV scale.**

| Model | S-B $\chi^2$ | df | S-B $\chi^2_{diff}$ [56] | $\Delta$df | CFI | $\Delta$CFI | RMSEA (90% CI) |
|---|---|---|---|---|---|---|---|
| 1) Equal form | 398.693 | 230 | -- | -- | .882 | -- | .055 (.046; .064) |
| 2) Equal factor loadings[a] | 416.919 | 243 | 18.223 | 13 | .885 | .003 | .055 (.046; .063) |
| 3) Equal indicator error variances for all 17 items[b] | 449.797 | 260 | 33.157* | 17 | .875 | -.010 | .055 (.046; .064 |
| Equal indicator error variances for . . . | | | | | | | |
| 3a) . . .16 items (without revised item 2[b]) | 449.318 | 259 | 32.812** | 16 | .874 | -.011 | .055 (.047; .064) |
| 3b) . . .16 items (without revised item 4[b]) | 444.496 | 259 | 27.585* | 16 | .877 | -.008 | .055 (.046; .063) |
| 3c) . . .16 items (without revised item 6[b]) | 440.547 | 259 | 23.370 | 16 | .880 | -.005 | .054 (.045; .063) |
| 3d) . . .16 items (without revised item 11[b]) | 449.318 | 259 | 32.713** | 16 | .874 | -.011 | .055 (.047; .064) |
| 3e) . . .16 items (without revised item 18[b]) | 438.350 | 259 | 21.052 | 16 | .881 | -.004 | .054 (.045; .062) |
| 3f) . . .16 items (without revised item 20[b]) | 449.719 | 259 | 33.201** | 16 | .874 | -.011 | .055 (.047; .064) |

[a] Comparison of Model 2 to Model 1

[b] Comparison of Models 3, 3a, 3b, 3c, 3d, 3e, and 3f to Model 2

*Note.* S-B $\chi^2$ = Satorra-Bentler scaled chi-square; df = degrees of freedom; S-B $\chi^2_{diff}$ = nested scaled $\chi^2$ difference requiring adjustment to chi-square change test; SRMR = standardized root mean square residual; CFI = comparative fit index; $\Delta$CFI = change in CFI for nested models; RMSEA = root mean square error of approximation; 90% CI = 90% confidence interval for RMSEA

\* $p < .05$

\*\* $p < .01$

recommended cutoff. With this in mind, subsequently more stringent ME/I tests were not conducted. See Table 5 for the fit indices.

To ascertain the specific sources of model misfit when error variances were constrained to equivalency, one-by-one the indicator error variances for the revised items were set free and comparisons to the baseline CFI of .882 were examined. When the error variance was freed one-at-a-time for Items 2, 4, 6, 11, 18, and 20 the $\Delta$CFI was -.011, -.008, -.005, -.011, -.004, and -.011, respectively. In sum, the error variances for three of six revised items were not invariant. See Table 4 for the model fit statistics. The error variance was larger in the control group for items 2 and 11 but not for item 20. Specifically, the error variance for item 2 was .93 in the control group and .88 in the treatment group. The error variance for item 11 was .99 in the control group and .88 in the treatment group. Finally, the error variance for item 20 was .59 in the control group and .82 in the treatment group. This suggests that the revised items 2 and 11 were more reliable (i.e. more variance was explained by the common latent construct than by any unknown source) than were the original items 2 and 11. However, the error variances for each of these three items were responsible for the majority of the overall variance extracted. The average variance extracted for the reduced length Mach IV with 17 items and four factors was nearly identical at 22.51% to the 22.11% for the control and the treatment groups, respectively.

## Study two discussion

Study Two examined the measurement invariance of originally worded and revised items in the Machiavellian IV scale [1] in an experimental framework. The scores on the two versions of the Mach IV showed equal form and equal factor loadings but unequal indicator error variances. Because invariance was detected at step three, the fourth and fifth tests of equal factor variances and equal factor covariances, respectively, were not conducted. The equal form of these two versions of the scale indicates that the pattern of covariances between the original and revised items is similar across the two groups (control and experimental groups). The result of equal factor loadings indicates that the items have similar variance emanating from

their focal underlying constructs (sub-scales of Machiavellianism) in both settings. Unequal indicator error variances suggest that the unexplained variance in the scale items is of a different magnitude in the two versions of the instrument.

The RMSEA and the SRMR were both within range of acceptability in all full-length and reduced-length iterations of the model in both groups, but the CFI suffered a bit in the scale's original entirety in the treatment group as well as in the multiple group analysis. The CFI for the final 17-item four-factor version of the Mach IV in the control and treatment groups was .914 and .856, respectively. These CFI values are not entirely out of line with previous efforts by other researchers. Previous CFA analyses of the Mach IV often also required the elimination of underperforming items and at least once, the revision of the wording for some items. Researchers found that the CFI ranged from .73 for 20 items using a four-point response scale [57], to .82 for 13 items [8], to .85 for all 20 items [58], and to .95 and .98 for ten-item versions of the Mach IV[9].

To further seek out the source of error invariance, six different single item error variances were freed one-by-one and the fit of these models were compared to the baseline model fit found in step one for equal form. It was found that the error variances for three of the six revised items (2, 11, and 20) were not invariant. Changes to the original items were: (a) the removal of an idiomatic expression " . . .carry more weight" in favor of " . . .be more influential" in item 2, (b) to correct both an archaic reference and an idiomatic expression with the complete rewrite of "Barnum was very wrong when he said there's a sucker born every minute" as "It is wrong to think that most people can be easily tricked" in Item 11, and (c) the removal of a strict gender reference such that "Most men are brave" was rewritten as "Most people are brave" in Item 20. These changes suggest that the error associated with these three items and therefore the reliability of the original and revised form was not equal. The alpha coefficient of reliability was .708 for the original 17 items and .687 for the revised scale of the same length.

Additionally, in an item-by-item comparison of scores in the control and treatment groups, three items had unequal variances by virtue of Levene's test. All were revised items. Item 6 about vicious streaks ($F = 4.234$, $p < .05$), item 11 about Barnum and suckers ($F = 4.221$, $p < .05$), and item 18 about men working hard ($F = 18.130$, $p < .001$) each had a larger spread of scores in the revised version than in the original version of the items. Item variances for other items in the control and treatment groups were not different. There were also significant differences in the means for those three items. Item 6 ($t = -4.234$, $df = 472.66$, $p < .001$, Cohen's $d = .381$) and item 18 ($t = -6.425$, $df = 466.15$, $p < .001$, $d = .580$) had higher means in the revised versions of the items and item 11 ($t = 7.992$, $df = 476.49$, $p < .001$, $d = .727$) had a higher mean in the original item. Item means for other items were not different. See Table 6 for the results for all items.

## General discussion

The fit of the 17-item four-factor model in single group and multiple group analysis was not altogether poor with the RMSEA consistently less than the cutoff of .06 and the SRMR in the single group analyses being less than .08. However, the CFI for both the single group and multiple group analyses fell slightly short at .88. The RMSEA indicates complex model misfit and its impressive performance on these data indicate that the pattern of covariances between the latent constructs (i.e. sub-scales) was reproduced effectively in the model. The CFI measures simple model misfit and it is no surprise that the weak factor loadings affected the ability to reproduce the model from these data. A separate SRMR is produced for both groups in ME/I tests as an indicator of each group's contribution to fit but an overall SRMR for the multiple group analysis is not obtainable when using Lisrel 8.8 software. Because changes in the CFI are

**Table 6. Tests of equality of variance and means for original versus revised items.**

| Item | Retained or discarded item? | Revised or original item? | Levene's test of equality of variance F | t-test of equality of means t | df | Cohen's d |
|------|------|------|------|------|------|------|
| 1. | retained | original | .573 | 1.734 | 481 | .157 |
| 2. | retained | revised | 1.946 | -.464 | 481 | .000 |
| 3. | retained | original | .018 | .076 | 481 | .005 |
| 4. | retained | revised | 1.116 | -.682 | 481 | .065 |
| 5. | retained | original | 3.853 | -.876 | 481 | .081 |
| 6. | retained | revised | 4.234* | -4.234*** | 472.66 | .381 |
| 7. | retained | original | .466 | -.044 | 481 | .007 |
| 8. | retained | original | .365 | -1.001 | 481 | .091 |
| 9. | discarded | revised | -- | -- | -- | -- |
| 10. | retained | original | .015 | .630 | 481 | .050 |
| 11. | retained | revised | 4.221* | 7.992*** | 476.49 | .727 |
| 12. | discarded | original | -- | -- | -- | -- |
| 13. | retained | original | .223 | 1.641 | 481 | .152 |
| 14. | retained | original | .488 | -.501 | 481 | .041 |
| 15. | retained | original | .141 | .459 | 481 | .041 |
| 16. | discarded | revised | -- | -- | -- | -- |
| 17. | retained | original | 1.617 | -.555 | 481 | .049 |
| 18. | retained | revised | 18.130*** | -6.425*** | 466.15 | .580 |
| 19. | retained | original | .058 | .335 | 481 | .031 |
| 20. | retained | revised | .041 | -1.868 | 481 | .173 |

*Note*. Non-integer degrees of freedom for some t-tests are an adjustment because of failing Levene's test of the homogeneity of variance

* $p < .05$

*** $p < .001$

the focus of analyses in ME/I tests [3], a marginally acceptable starting value for the groups may have affected the results of subsequent tests on the data. In sum, the results of revising items in the Mach IV did not produce great change in model fit. With these results and those of other CFA tests of the Mach IV it may be time to move on to other more recently developed measures of Machiavellianism. The wildly fluctuating number of factors resulting from previous work, the oddly changing loading of items on different factors in different published studies, and our own ME/I tests suggest that the problems with the Mach IV might be insurmountable. We encourage other researchers to continue their development of alternative measures of Machiavellianism and support a move toward the measurement of actual Machiavellian behavior using multiple sources of information which will surely aid in the collection of validity evidence for the construct.

The results of research on the measurement of Machiavellianism appears to depend greatly on the intended audience for the instrument. However, much if not most initial psychometric analyses of self-report inventories is conducted using college students. These students are rarely the intended respondents of such scales and this should be noted as a limitation of the current study. Sample-specific instruments do exist which do not make use of college students in the scale development phase such as the Kiddie Mach which was developed for children [1]. With the passage of time the language of items in scales periodically requires updating because of demographic changes in typical respondents. Efforts at changing the wording of some of the items in the Mach IV in the current study produced only minor change in the fit of the models

to the data. The measurement of Machiavellianism is likely to be more complex than originally envisioned in 1970 [1]. Given the inclusion of Machiavellianism in the Dark Triad [5], measurement research on the construct is likely to proliferate in the future.

## Strengths

Some strengths of the current research are its experimental design and the use of stringent measurement tests associated with CFA and measurement invariance. The age of the respondents was also a strength given the propensity of psychological researchers to use undergraduate students and the main premise of this research that the lexicon of students today is likely different from that of students approximately 50 years ago when the scale was developed. Another strength is the use of CFA-based ME/I tests [2] given that less stringent guidelines for ME/I steps and tests have been advocated by others [59,60].

## Future research

Researchers may want to consider pursuing some areas of related research. First, to gather evidence of the validity of the revisions to the Mach IV proposed here, they might consider examining the impact of our revised Mach IV scale in its relationship with other constructs. Second, they might consider conducting an item response theory (IRT) analysis on the revised items in the instrument. Such an analysis would help ascertain item level discrimination and difficulty as well as where the most item information is found on the response scale for the revised items of the Mach IV. The dependable, consistent measurement of any construct is paramount and an IRT-based analysis of item-level reliability would likely help with the effective measurement of Machiavellianism.

Another promising area of research is to extend this type of framework by revising problematically worded items in other scales. For example, the Protestant Work Ethic instrument [61] and the Conscientiousness scale from the International Personality Item Pool [62] have some potential problems with wording. The PWE items potentially suffer from non-gender-neutral references such as: "The self-made man is likely to be more ethical than the man born to wealth" and "Any man who is able and willing to work hard has a good chance of succeeding." To a female-dominated sample of college student respondents these items may be conceptualized differently than they are by male respondents. The IPIP's Conscientiousness scale potentially suffers from the use of archaic phrasing. One item reads as "I shirk my duties" (reverse scored). The use of the word "shirk" peaked in 1916 [63] and may be completely unfamiliar to a sample of today's college students. Scales developed almost 50 years ago like the PWE, or even as recent as 20 years ago as with the IPIP, may suffer from the sort of archaic references, non-gender neutral inferences, and idiomatic expressions examined in this study and are therefore worthy of examination in a ME/I framework as well.

## Author Contributions

**Conceptualization:** Brian K. Miller.

**Formal analysis:** Brian K. Miller.

**Writing – original draft:** Brian K. Miller, Kay Nicols.

**Writing – review & editing:** Brian K. Miller, Kay Nicols.

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
