## [Decision Letter · Decision Letter 0]

27 Jun 2019

PONE-D-19-15176

Measurement invariance tests of revisions to archaically worded items in the Mach IV scale

PLOS ONE

Dear Dr. Miller,

Thank you for submitting your manuscript to PLOS ONE. After careful consideration, we feel that it has merit but does not fully meet PLOS ONE’s publication criteria as it currently stands. Therefore, we invite you to submit a revised version of the manuscript that addresses the points raised during the review process.

Please, see the comments of three Reviewers appended at the bottom of this letter. I am sorry that the two-lines comment provided by Reviewer #2 are quite useless. In contrast, Reviewer #1 has offered you constructive feedback, which I think might contribute to improve the presentation of your study. Because this might be considered as a major review, please notice that a resubmission will require an additional round of reviews, and that the final outcome of the process cannot be predicted at this point. If you decide to resubmit a revised version of your manuscript, please provide either a proper answer or rebuttal to each of the suggestions that were raised by the Reviewers.

We would appreciate receiving your revised manuscript by Aug 11 2019 11:59PM. To enhance the reproducibility of your results, we recommend that if applicable you deposit your laboratory protocols in protocols.io, where a protocol can be assigned its own identifier (DOI) such that it can be cited independently in the future. For instructions see: http://journals.plos.org/plosone/s/submission-guidelines#loc-laboratory-protocols

We look forward to receiving your revised manuscript.

Kind regards,

Angel Blanch, Ph.D.

Academic Editor

PLOS ONE

Journal Requirements:

3. Please address the following queries related to the Mach IV scale modified in the current study: 1) If the questionnaire is licensed, do you have permission to use the licensed questionnaire for the purposes of the study? 2) As the questionnaire has been published previously, please state whether you have permission to reprint items of the published questionnaire under a CC-BY license?

Please provide additional details regarding participant consent. In the ethics statement in the Methods and online submission information, please ensure that you have specified (1) whether consent was informed and (2) how the verbal consent was documented and witnessed. If your study included minors, state whether you obtained consent from parents or guardians. If the need for consent was waived by the ethics committee, please include this information.”

Reviewers' comments:

Reviewer's Responses to Questions

**Comments to the Author**

1. Is the manuscript technically sound, and do the data support the conclusions?

Reviewer #1: Partly

Reviewer #2: Yes

2. Has the statistical analysis been performed appropriately and rigorously? 

Reviewer #1: Yes

Reviewer #2: Yes

3. Have the authors made all data underlying the findings in their manuscript fully available?

Reviewer #1: Yes

Reviewer #2: Yes

4. Is the manuscript presented in an intelligible fashion and written in standard English?

Reviewer #1: Yes

Reviewer #2: Yes

5. Review Comments to the Author

Reviewer #1: General comment:

The authors conduct exploratory and measurement invariance analyses of a scale which is intended to measure the construct Machiavellianism. More specifically, they examine the impact of a rewording of some of the items on the structure and measurement properties of the scale.

The method of analysis and the rationale underlying it are described clearly. However, I think that some of the conclusions which are drawn from the analysis are debatable. For instance, an initial step of testing weak measurement invariance, results in a mediocre model fit. Given this (mis)fit, the subsequently drawn conclusions from the model comparisons do not necessarily seem stringent (see comments below). Moreover, it should be stated that the reported fit-indices are likely to be biased upward, given that preliminary model selection was already applied (that is, a preliminary elimination of “bad” items was conducted). Finally, I also think that the usage of a 4-6-factor solution for a test comprised of less than 20 items is problematic (see comments below) and should be discussed appropriately.

Specific comments:

Introduction: The paper would benefit from providing some brief review of the theory underlying the construct of Machiavellianism. In addition, providing the content/wording of all items would be helpful in understanding the proposed factor structure.

34: Factor indeterminacy is a general problem prominent in any FA model.

35-38: This suggests large variation between different (study) populations.

51:53: This could be tested via factorial invariance. Are there any corresponding results in the literature. Why does the current study not include such a test of invariance across gender?

The aim seems to be to optimize the MACH for student populations, as a lot of the presented arguments are tied to changes in student populations, i.e. a change in the proportion of female and international students. This should be mentioned as a limitation because I don't think that students are the primary focus for the use of a diagnostic instrument for Machiavellianism.

115: I would avoid the usage of cut-off scores like .3. On the one hand, they are somewhat arbitrary. But more importantly: It can be shown that even small loadings (below .3) can have a large impact on the inference of the factor scores when diagnosing test takers (see e.g. Jordan & Spiess, 2019).

133: It is correctly stated that alpha only provides a lower bound for the overall reliability. However, even if alpha were sufficiently high, it would - in my opinion – still be of limited use because its reference point is a formally defined "overall" true score which in this case is a mixture of multiple (6) dimensions.

141: There are more reliable approaches to the determination of the number of factors like e.g. Horn's parallel analysis (PA) or the more recently developed “deterministic counterpart” based on random matrix theory (see Dobriban & Owen, 2019). The authors already mentioned large variations of the number of extracted factors across different studies. Hence, I wonder if some part of the variation could be explained by "suboptimal" extraction criteria. In any case, I would suggest the use of PA to determine the number of factors.

162: The authors should expand their argument on negative loadings. In general, negative loadings are not a problem unless they contradict the factor label (i.e. an item in an intelligence test with negative loadings would contradict the label "IQ" as test takers with lower IQ would score higher on the item).

233: The examination of normality is not necessary. We already know that normality can't hold due to the discrete (1 to 7) format of the responses.

246-250: How was the CFA specified? By usage of the exact loadings from the EFA or by treating loadings below .3 as zero? In addition, heywood cases should not arise under ML-estimation. If the ML method did not converge (to which I think the heywood case refers), then this points to problems in the specification of the model.

263: I do not understand what is meant by "the covariances of ... were unidentified". Does this mean that the estimated factor covariance matrix was not positive definite?

264: It seems problematic to use such a few number of items per factor. Measuring a factor by 2-5 items is tantamount to produce scorings with low reliability.

275: "unidentified": It would be helpful to distinguish an unindentifiable case from a case, wherein the numerical optimization did not converge and/or provided inadequate estimates (e.g. negative variances). Which type are the authors referring to?

308: The problem with the approach for determining various sorts of measurement invariance is that the baseline model does not provide a good fit. Hence, although it seems that given the baseline model, further restrictions are possible without substantially lowering the CFI, the fit of the baseline model itself is somewhat questionable. Given these doubts, I think that the subsequent comparison of error variances is problematic as it relies on a good overall model fit (and not just on a good fit relative to a model with mediocre fit).

372: This line of reasoning should be strengthened by computing model based reliability estimates. In addition, to demonstrate an effect of item wording, I would recommend to append the paper with a test of the treatment effect. That is, are there any difference in the distribution of the responses to an item between treatment and control group?

395: In the general discussion, critique of the MACH is mentioned. I think it would be important to add and discuss the following topics:

- How can we account for (or interpret) the large variations (e.g. number of factors; number of retained items) across studies?

- If the wording of items has an impact on the reliability (or potentially even on the factor structure) of the scale, then this leaves "us" with measurement devices (self-report questionnaires) which are very fragile. Hence, I think a point could be made here in favor of moving towards other (perhaps more costly) measurement devices (e.g. actual observation of behavior; using multiple sources of information etc.).

- Given the relatively large number of underlying factors (4-6), the researcher/practitioner has the option to

a) either compute an inhomogeneous overall score which refers to a (difficult to interpret) mixture of (4-6) constructs

or

b) to compute dimension specific scores (4-6).

However, choosing b) basically boils down to measure a latent construct by only a few items. Hence, the scores are highly unreliable.

- Related to the previous point, but broadening the scope: A topic of central importance in the analysis of the replication crisis in psychology referred to the role of the measurement error (see e.g. Loken & Gelman, 2013), i.e. measurements in psychology are in general rather noisy. I think that by using such short (on average 4-5 items per factor) subscales, classical test theory would predict unreliable, noisy measurements. Thus, their subsequent usage might entail all of the problems which were discussed within the context of the low replicability of psychological science. Hence, I regard this as an additional argument against the usage of the scale.

References:

Dobriban, E. & Owen, A. B. (2019). Deterministic parallel analysis: an improved method for selecting factors and principal components. J. R. Stat. Soc. B, 81: 163-183. doi:10.1111/rssb.12301.

Jordan, P. & Spiess, M. (2019). Rethinking the interpretation of item discrimination and factor loadings, Educational and Psychological Measurement.

Loken, E., & Gelman, A. (2017). Measurement error and the replication crisis. Science, 355(6325), 584–585.

Review is also available in the attached file.

Reviewer #2: The manuscript offers all the statistical information; also, an exhaustive and rigorous analysis process has been made. However, a more consistent theoretical introduction is lacking.

6. PLOS authors have the option to publish the peer review history of their article (what does this mean?). If published, this will include your full peer review and any attached files.

Reviewer #1: No

Reviewer #2: No

---

## [Author Response · Author response to Decision Letter 0]

1 Aug 2019

Editorial Comments: 

1. When submitting your revision, we need you to address these additional requirements. Please ensure that your manuscript meets PLOS ONE's style requirements, including those for file naming. 

Authors’ response: Thank you. We have done so. 

Authors’ response: Thank you. We will be happy to do so.

3. Please address the following queries related to the Mach IV scale modified in the current study: 1) If the questionnaire is licensed, do you have permission to use the licensed questionnaire for the purposes of the study? 2) As the questionnaire has been published previously, please state whether you have permission to reprint items of the published questionnaire under a CC-BY license?

Authors’ response: To our knowledge the instrument is in the public domain. It has been published in its entirety many, many times by numerous researchers and internet web sites. For example, Rauthmann (2013) published the entire scale in his article using IRT to analyze the Mach IV. The web site known as “A conscious rethink” published the entire scale at https://www.aconsciousrethink.com/6299/machiavellian-scale-test/ , for example. Hundreds of sites and articles have republished it in its entirety.

4. Please provide additional details regarding participant consent. In the ethics statement in the Methods and online submission information, please ensure that you have specified (1) whether consent was informed and (2) how the verbal consent was documented and witnessed. If your study included minors, state whether you obtained consent from parents or guardians. If the need for consent was waived by the ethics committee, please include this information.”

Authors’ response: The need for consent was waived by our university’s Institutional Review Board who approved the study as exempt. Nevertheless, consent was indeed informed and acknowledged by respondents via signed consent forms. The forms were collected separately from the actual paper-and-pencil surveys so as to maintain the anonymity of respondents and were used to record participation by the professor from whose class respondents were voluntarily solicited so as to award some very, very minor extra credit in the course. 

 

Reviewer #2 Comments: 

1. The manuscript offers all the statistical information; also, an exhaustive and rigorous analysis process has been made. However, a more consistent theoretical introduction is lacking.

Authors' reply: We thank the reviewer for their comment on our rigor. We take great pride in our statistical prowess and are happy that the reviewer noticed. Given the broad audience for PlosONE we were remiss in our duties with the original submission regarding the theoretical foundation. The other reviewer made the same suggestion. Thus, we have included an entirely new introductory paragraph starting on line 27 and slightly altered the lead sentence in the now-second paragraph as follows: 

“Machiavellianism is the predisposition to manipulate interpersonal relationships with guile, opportunism, and deceit. Research on this set of traits began in earnest with the development of a self-report inventory [1] based upon the lead character in Niccolo’ Machiavelli’s sixteenth century novel “The Prince”. The title character lacked morality and empathy and had ample distrust of others. The scale authors [1] painstakingly developed items designed to measure the character’s tendency toward amorality, to have negative views of human nature, and to employ interpersonal tactics designed for personal gain at the expense of others [1]. The result was the Mach IV which is a 20-item scale comprised of three subscales: morals, views, and tactics. This instrument has become the most frequently used scale to measure Machiavellianism [1]. Interest in Machiavellianism was further with its inclusion as one of the three traits comprising the relatively new Dark Triad [5]. The other two traits are…” We hope that is will suffice but would be happy to add more should the reviewer or the editor require it. 

 

Reviewer #1 Comments:

The authors conduct exploratory and measurement invariance analyses of a scale which is intended to measure the construct Machiavellianism. More specifically, they examine the impact of a rewording of some of the items on the structure and measurement properties of the scale.

1. The method of analysis and the rationale underlying it are described clearly. However, I think that some of the conclusions which are drawn from the analysis are debatable. For instance, an initial step of testing weak measurement invariance, results in a mediocre model fit. Given this (mis)fit, the subsequently drawn conclusions from the model comparisons do not necessarily seem stringent (see comments below). 

Authors’ reply: We must admit that we see our paper as a sort of latch-ditch effort to save the Mach IV scale. As we note in our paper, the number of factors extracted by previous authors has varied wildly and the need to eliminate many items to achieve model fit by others is also commonplace. We want to be polite to Christie and Geis, however. With this sort of professional courtesy in mind we treaded lightly on our view of the scale. We thought that by revising some of the problematically worded items we could perhaps save the instrument from its current precipitous decline in usage due in part to other newly developed scales that we note in the paper and the well documented problems with the Mach IV scale. As it turns out, our revised scale shows some differences in the error variances leading one to conclude that its reliability differs from the original. We address the reviewer’s points one-by-one below and thank the reviewer for their insightful comments and suggestions. 

2. Moreover, it should be stated that the reported fit-indices are likely to be biased upward, given that preliminary model selection was already applied (that is, a preliminary elimination of “bad” items was conducted). 

Authors’ reply: We agree and have added the following text starting on line 294: “However, because of sequential item eliminaton these fit statistics are likely biased upward and subsequent analysis based on this shortened scale should be viewed with caution given the changes to the instrument and its factor structure that were required to achieve model fit.“ We do ask the reviewer to note that previous efforts at factor analysis of the inventory required very similar item elimination and factor reductions. In essence our factor analytic results are not very different from those of other researchers on the Mach IV instrument. 

3. Finally, I also think that the usage of a 4-6-factor solution for a test comprised of less than 20 items is problematic (see comments below) and should be discussed appropriately.

Authors’ reply: We apologize for not being more transparent in our review of other’s factor analytic work on the Mach IV. The reviewer will hopefully note that Table 1 contains ample previous evidence of EFA/PCA analyses of the Mach IV that found between 3 and 7 factors for the 20 items. This sort of factor indeterminancy seems to be the norm in analysis of the Mach IV. In fact, Christie and Geis (1970) recommended a three-factor solution with one factor comprised of only two items. Williams et al.’s (1975) multi-factor solution had two factors with only two items ewach loading on them. Kuo and Marsella’s (1977) U.S. sample allowed six items to cross-load on at least two different factors. On page 6 the original text mentions the following: “This variety of factor solutions and very different loadings of items on factors in the Mach IV instrument suggests that further factor analyses are in order to build an understanding of the underlying factor structure of the Mach IV.“ However, we agree with the reviewer that this is not likely to be enough. We have amended the last sentence in the paragraph ending on page 19 to read as follows: "It is noteworthy that two of the three items discarded were revised items and that this factor structure is not dissimilar to that of previous factor analyses of of the Mach IV instrument which vary wildly from three to seven factors on items ranging in number from 10 to 20.“ Please also note that the original text on page 24 reads as follows: “Previous CFA analyses of the Mach IV often also required the elimination of underperforming items and at least once, the revision of the wording for some items. Researchers found that the CFI ranged from .73 for 20 items using a four-point response scale [57], to .82 for 13 items [8], to .85 for all 20 items [58], and to .95 and .98 for ten-item versions of the Mach IV[9].“ We hope the reviewer agrees that we have adequately addressed this very valid concern and that our models are not dissimilar to those found in previous research on the Mach IV. 

Specific comments:

4. Introduction: The paper would benefit from providing some brief review of the theory underlying the construct of Machiavellianism.

Authors’ reply: The other reviewer also made this suggestion. Thus, we have included an entirely new introductory paragraph starting on line 27, which reads as follows: “Machiavellianism is the predisposition to manipulate interpersonal relationships with guile, opportunism, and deceit. Research on this set of traits began in earnest with the development of a self-report inventory [1] based upon the lead character in Niccolo’ Machiavelli’s sixteenth century novel “The Prince”. The title character lacked morality and empathy and had ample distrust of others. The scale authors [1] painstakingly developed items designed to measure the character’s tendency toward amorality, to have negative views of human nature, and to employ interpersonal tactics designed for personal gain at the expense of others [1]. The result was the Mach IV which is a 20-item scale comprised of three subscales: morals, views, and tactics. This instrument has become the most frequently used scale to measure Machiavellianism [1]. Interest in Machiavellianism was further with its inclusion as one of the three traits comprising the relatively new Dark Triad [5]. The other two traits are...“ 

We hope that this helps alleviate the reviewer’s concerns but would be delighted to add more if this is not sufficient. 

5. In addition, providing the content/wording of all items would be helpful in understanding the proposed factor structure.

Authors’ reply: We suggest that the items not being revised are of less concern than those which did undergo some revision in our study. The original and revised items are included in Table 3 and the items not being revised are nevertheless still important but only a minor part of our analysis. The entire scale has been republished in its entirey scores, if not hundreds, of times and is available eslewhere. We are fans of Rauthmann’s (2013) IRT paper published in the Journal of Personality Assessment which contains the full scale. 

6. Line 34: Factor indeterminacy is a general problem prominent in any FA model. 

Authors’ response: We agree and have added the following to line 48: “…are not unlike other lengthy scales with obliquely related subscales...“ Thank you for this suggestion. 

7. Lines 35-38: This suggests large variation between different (study) populations. 

Authors’ response: Thank you. We have added “…due in part to the diversity of populations to which it has been administered“ to those lines. This is important to note. 

8. Lines 51:53: This could be tested via factorial invariance. Are there any corresponding results in the literature. Why does the current study not include such a test of invariance across gender?

Authors’ response: To our knowledge this has not been studied yet. However, the current study is about revised versus orginal items, not males versus females per se. However, part of our argument is that some modern women may object to non-gender neutral items. As the reviewer suggests, the next step is to test whether males and females conceptualize of our revised items differently in tests of measurement invariance. It is, perhaps, a study that we will do in the future. We thank the reviewer for this query but it is beyond the scope of the current study due in part to the sample size requirements associated with statistical power and the very specific focus of the current paper. 

9. The aim seems to be to optimize the MACH for student populations, as a lot of the presented arguments are tied to changes in student populations, i.e. a change in the proportion of female and international students. This should be mentioned as a limitation because I don't think that students are the primary focus for the use of a diagnostic instrument for Machiavellianism.

Authors’ response: We understand this point very clearly. It is true that our study focuses on student respondents and the Mach IV was not designed to measure Machiavellianism only in students. However, our focus on students as respondents is because of their proliferate usage in the initial scale development stage of many if not most self-report inventories. To that end we have added the following to pages 26-27: “However, much if not most, initial psychometric analyses of self-report inventories is conducted using college students. These students are rarely the intended respondents of such scales and this should be noted as a limitation of the current study. Sample-specific instruments do exist which do not make use of college students in the scale development phase such...“ Thank you very much for this reminder. 

10. Line 115: I would avoid the usage of cut-off scores like .3. On the one hand, they are somewhat arbitrary. But more importantly: It can be shown that even small loadings (below .3) can have a large impact on the inference of the factor scores when diagnosing test takers (see e.g. Jordan & Spiess, 2019).

Authors’ response: We apologize for not highlighting more strongly the arbitrariness of such cutoffs in our original submission. We did this because we needed to decide in some manner which items to force onto which factors in the CFA analysis that followed. The CFA analysis did not allow cross-loadings as is the norm. Using a stricter cutoff of .4 would have resulted in six of the 20 original items not loading at all on any factor. A more lenient cutoff of .2 would have forced us to include six items with cross-loadings in our CFA analysis which we wanted to avoid very much. Because our intent was to confirm the EFA factor structure with CFA in the next step of our study, we had to use some cutoffs. We hope that our current caution about using cutoffs is enough to satisfy the reviewer. Howver, we would be delighted to include any additional information should it be absolutely required. We thank the reviewer for this suggestion. 

11. Line 133: It is correctly stated that alpha only provides a lower bound for the overall reliability. However, even if alpha were sufficiently high, it would - in my opinion – still be of limited use because its reference point is a formally defined "overall" true score which in this case is a mixture of multiple (6) dimensions.

Authors’ response: We agree and have added the following to the middle of the sentence of note on page 12: “…the difficulty of estimating a baseline true score resulting from scale...“ Thank you for this reminder. 

12. Line 141: There are more reliable approaches to the determination of the number of factors like e.g. Horn's parallel analysis (PA) or the more recently developed “deterministic counterpart” based on random matrix theory (see Dobriban & Owen, 2019). The authors already mentioned large variations of the number of extracted factors across different studies. Hence, I wonder if some part of the variation could be explained by "suboptimal" extraction criteria. In any case, I would suggest the use of PA to determine the number of factors. 

Authors’ response: We thank the reviewer. After reading this suggestion, we ran parallel analysis in JASP software, which easily conducts EFA. The parallel analysis also resulted in a six-factor solution. We have added some text to reflect this and we thank the reviewer for recommending it to us. 

13. Line 162: The authors should expand their argument on negative loadings. In general, negative loadings are not a problem unless they contradict the factor label (i.e. an item in an intelligence test with negative loadings would contradict the label "IQ" as test takers with lower IQ would score higher on the item).

Authors’ response: Thank you for this suggestion that we expand upon our findings/assertions. These negative loading do indeed contradict the factor labels. We have added the following two sentences to pages 12-13 in the hopes of further clarifying this issues for potential future readers: “That is, items with negative loadings tend to measure the opposite end of the Machiavellianism spectrum and when summed with positively loading items the overall scale score is closer to the middle range than either the low or high ranges. Therefore, the Mach IV is unlikely to be effective in discriminating low or high Machs from mid-range Machs.“ We hope that this is what the reviewer had in mind but would be delighted to exand upon this issue further if another revise and resubmit is graciously offered or required. 

14. Line 233: The examination of normality is not necessary. We already know that normality can't hold due to the discrete (1 to 7) format of the responses. 

Authors’ response: We will politely disagree here. The two issues that determine whether the default maximum likelihood estimation procedure can be used are coarseness of measurement and multivariate normality. Scales with as few as four points have been determined to not be so course as to require weighted least squares estimation. Our response scale was seven points. However, the data were not multivariate normal. The first step in the assessment of item normality is the determination of univariate normality. If and only if the data are not univariate normal can one correctly assume that the data are not multivariate normal. Our data were univariate normal according to commonly accepted cutoffs for skewness and kurtosis. Therefore we had to then test for multivariate normality. If the items had not been univariate normal we would not have had to calculate Mardia’s coefficient because univariate normality is a necessary but not sufficient condition for multivariate normality. Our data were not multivariate normal so because of the lack of coarseness of measurement in the items but the fact that the data were not multivariate normal we used the Satorra-Bentler corrections. If the data had been more coarsely measured but multivariate normal we would have used weighted least squares estimation. We think coverage of how we determined the appropriate estimator function is critical to our analyses and is sorely lacking from much of the research that we read. We hope the reviewer will allow us to keep this information in the manuscript but will relent and delete it if absolutely required. 

15. Lines 246-250: How was the CFA specified? By usage of the exact loadings from the EFA or by treating loadings below .3 as zero? In addition, heywood cases should not arise under ML-estimation. If the ML method did not converge (to which I think the heywood case refers), then this points to problems in the specification of the model.

Authors’ response: We regret the omission of details regarding the CFA model specification. We apologize. Factor loadings less than .3 from the EFA test were indeed treated as zero. To use the exact loadings of the EFA model in a cross-loaded CFA test would likely have resulted in an overly complicated model with near perfect fit. Our goal, like that of most CFA tests was to find as parsimonious of a model of the data as possible. To address our omission we have added the following paragraph to page 15: “In the CFA model tests that follow, the factor structure arising from the previous EFA test was imposed on the factor structure with items loading less than .30 in the EFA being forced to zero in the CFA. The goal was to use as parsimonious of a model of the data as possible. Therefore error terms was not allowed to correlate and items were not allowed to cross-load on multiple factors.“ Please also note that in the original version of our paper the following text was found in the subsection on "Preliminary single group analysis" (now on page 17) and reads as follows: "In the preliminary analysis with the control group examining the six-factor structure of the 20-item Mach IV found via EFA in Study One."

As to the Heywood cases being evidence of non-convergence, the reviewer is correct and the fact that they occured is what forced us to rework the specification of our model by eliminating items and factors. Technically, Heywood cases are not uncommon in ML estimation and certainly arise more often in ML than in both ordinary least squares or generalized least squares estimation procedures. 

16. Line 263: I do not understand what is meant by "the covariances of ... were unidentified". Does this mean that the estimated factor covariance matrix was not positive definite? 

Authors’ response: Yes, it does. The term “unidentified” is part of the Lisrel error message we received. Another way of describing such an error is to say that the relationship between the two variables is so unstable so as to not allow for the estimation of one best covariance. We have added some verbiage to that effect on page 18: “indicating that the covariance was unstable and not likely to result in any particular best estimate of the data.“ We hope this clarifies things a bit for the reviewer. 

17. Line 264: It seems problematic to use such a few number of items per factor. Measuring a factor by 2-5 items is tantamount to produce scorings with low reliability.

Authors’ response: We agree and note in Table 1 the proliferation of such items-per-factor including the original scale by Christie and Geis. This is just one more of the many problems with this scale. 

18. Line 275: "unidentified": It would be helpful to distinguish an unindentifiable case from a case, wherein the numerical optimization did not converge and/or provided inadequate estimates (e.g. negative variances). Which type are the authors referring to?

Authors’ response: This is a typographical error of sorts on our part. We have changed the verbiage to refer to the problem with item 9 in the 18-item CFA model as having “negative error variance”. We thank the reviewer for bringing this to our attention. 

19. Line 308: The problem with the approach for determining various sorts of measurement invariance is that the baseline model does not provide a good fit. Hence, although it seems that given the baseline model, further restrictions are possible without substantially lowering the CFI, the fit of the baseline model itself is somewhat questionable. Given these doubts, I think that the subsequent comparison of error variances is problematic as it relies on a good overall model fit (and not just on a good fit relative to a model with mediocre fit).

Authors’ response: We thank the reviewer for this keen insight. As part of our response to the reviewer’s general comment #2, we address this here. Previous researchers have had to engage in all sorts of machinations to examine the model fit of the Mach IV. Ours is no different. The scale is a bit of a mess. The comparison of error variances is just the next step in Brown’s (2006) bottom-up approach to measurement invariance tests as codified by Cheung and Rensvold (1999; 2002). It is true that our changes in model fit from step to step are comparing one bad apple to another, but we worked with what we had. Our data collection process was tightly controlled and we trust the responses to our surveys. Even when revised, the scale still has problems. We wish it were different. We really do. 

20. Line 372: This line of reasoning should be strengthened by computing model based reliability estimates. In addition, to demonstrate an effect of item wording, I would recommend to append the paper with a test of the treatment effect. That is, are there any difference in the distribution of the responses to an item between treatment and control group?

Authors’ response: In our original manuscript we have an examination of item-by-item error variances immediately following Table 5 that is quite lengthy. Overall, the 17-item scale had similar coefficient alphas. We have added the following to the section on results for study two: "The alpha coefficient of reliability was .708 for the original 17 items and .687 for the revised scale of the same length." We hope that this is what the reviewer had in mind. We also conducted independent samples t-tests on the items as well as Levene’s test of the homogeneity of variance between the control and treatment groups. The results are rather brief as few differences exist and we decided to add a paragraph detailing them to the end of the study two results section. It reads as follows: "Additionally, in an item-by-item comparison of scores in the control and treatment groups, only two items had unequal variances by virtue of Levene’s test. Item 11 about Barnum and suckers resulted in F = 4.86, p < .05 and item 18 about men and hard work resulted in F = 20.985 (p < .01). For these two items the spread of scores was significantly larger in the treatment group than in the control group. Regarding mean differences in scores on the items, item 6 about vicious streaks resulted in t = -4.19 (p < .001), item 11 resulted in t = 7.76 (p < .001) and item 18 resulted in t = -6.60 (p < .001). For items 6 and 18, the mean score was higher in the treatment group but the mean for item 11 was higher in the control group. All in all, there were significant differences in either the distribution of scores or the mean scores on three items, each of which was a revised item." We hope that these rather brief item comparisons are acceptable to the reviewer and we thank the reviewer for this very important suggestion. 

21. Line 395: In the general discussion, critique of the MACH is mentioned. I think it would be important to add and discuss the following topics:

Authors’ response: Thank you. See below. 

21A. - How can we account for (or interpret) the large variations (e.g. number of factors; number of retained items) across studies?

Authors’ resonse: Hmmm. This is a good question. We assume that it is largely but not completely because of idiosyncracies in different groups of respondents. Our analysis shows that the item wording had only a minimal effect on model fit. The problem surely lies at the intersection of the items and survey respondents. That is, different respondent interpret different items differently. That, of course, is a problem with the original scale not being particularly great. It was a giant leap forward in measuring dark personality traits 50 years ago, but modern psychometric analyses have shown that it simply may be time to retire this scale, as we noted at the end of the first paragraph in the general discussion section. As we mentioned in our response to the reviewer’s general comment #1 we want to be polite to people in general and to our colleagues engaged in this sort of research in particular, so we have added the following to this paragraph based upon the reviewer’s query: “The wildly fluctuating number of factors resulting from previous work, the oddly changing loading of items on different factors in different published studies, and our own ME/I tests suggest that the problems with the Mach IV might be insurmountable.“

21B. - If the wording of items has an impact on the reliability (or potentially even on the factor structure) of the scale, then this leaves "us" with measurement devices (self-report questionnaires) which are very fragile. Hence, I think a point could be made here in favor of moving towards other (perhaps more costly) measurement devices (e.g. actual observation of behavior; using multiple sources of information etc.). 

Authors’ response: We agree and have added the following to the aforementioned paragraph above: “We encourage other researchers to continue their development of alternative measures of Machiavellianism and support a move toward the measurement of actual Machiavellian behavior using multiple sources of information which will surely aid in the collection of validity evidence for the construct.“

21C. - Given the relatively large number of underlying factors (4-6), the researcher/practitioner has the option to a) either compute an inhomogeneous overall score which refers to a (difficult to interpret) mixture of (4-6) constructs, or b) to compute dimension specific scores (4-6). However, choosing b) basically boils down to measure a latent construct by only a few items. Hence, the scores are highly unreliable. 

Authors’ response: We agree and suggest that for those administering this in a diagnostic effort at understanding Machiavellian facets the use of subscale scores is best. However, as the reviewer notes, these very short measures are inherently unreliable. In contrast, simply summing the scores on the subscales for an overall measure of Machiavellianism is painting with a broad brush. Additionally, as the reviewer notes adding heterogenous subscales to magically achieve an acceptable reliability is problematic in its own right. 

21D. - Related to the previous point, but broadening the scope: A topic of central importance in the analysis of the replication crisis in psychology referred to the role of the measurement error (see e.g. Loken & Gelman, 2013), i.e. measurements in psychology are in general rather noisy. I think that by using such short (on average 4-5 items per factor) subscales, classical test theory would predict unreliable, noisy measurements. Thus, their subsequent usage might entail all of the problems which were discussed within the context of the low replicability of psychological science. Hence, I regard this as an additional argument against the usage of the scale. 

Authors’ response: We agree and as noted previously we want to be polite in our mild condemnation of the scale. We have added verbiage aluded to that which hopefully is strong enough for the reviewer. Overall, we hope that we have adequately addressed the reviewer's points and made appropriate changes to the paper where needed. Our paper is much better because of the reviewer and we hope the reviewer and editor agree. Thanks!

References:

Dobriban, E. & Owen, A. B. (2019). Deterministic parallel analysis: an improved method for selecting factors and principal components. J. R. Stat. Soc. B, 81: 163-183. doi:10.1111/rssb.12301.

Jordan, P. & Spiess, M. (2019). Rethinking the interpretation of item discrimination and factor loadings, Educational and Psychological Measurement.

Loken, E., & Gelman, A. (2017). Measurement error and the replication crisis. Science, 355(6325), 584–585.

---

## [Decision Letter · Decision Letter 1]

29 Aug 2019

[EXSCINDED]

PONE-D-19-15176R1

Measurement invariance tests of revisions to archaically worded items in the Mach IV scale

PLOS ONE

Dear Dr. Miller,

Thank you for submitting your manuscript to PLOS ONE. After careful consideration, we feel that it has merit but does not fully meet PLOS ONE’s publication criteria as it currently stands. Therefore, we invite you to submit a revised version of the manuscript that addresses the points raised during the review process.

We would appreciate receiving your revised manuscript by Oct 13 2019 11:59PM. To enhance the reproducibility of your results, we recommend that if applicable you deposit your laboratory protocols in protocols.io, where a protocol can be assigned its own identifier (DOI) such that it can be cited independently in the future. For instructions see: http://journals.plos.org/plosone/s/submission-guidelines#loc-laboratory-protocols

We look forward to receiving your revised manuscript.

Kind regards,

Angel Blanch, Ph.D.

Academic Editor

PLOS ONE

Reviewers' comments:

Reviewer's Responses to Questions

**Comments to the Author**

1. If the authors have adequately addressed your comments raised in a previous round of review and you feel that this manuscript is now acceptable for publication, you may indicate that here to bypass the “Comments to the Author” section, enter your conflict of interest statement in the “Confidential to Editor” section, and submit your "Accept" recommendation.

Reviewer #1: (No Response)

2. Is the manuscript technically sound, and do the data support the conclusions?

Reviewer #1: Yes

3. Has the statistical analysis been performed appropriately and rigorously? 

Reviewer #1: Yes

4. Have the authors made all data underlying the findings in their manuscript fully available?

Reviewer #1: No

5. Is the manuscript presented in an intelligible fashion and written in standard English?

Reviewer #1: Yes

6. Review Comments to the Author

Reviewer #1: The authors have addressed most of the points which were raised. They also provided convincing arguments for their approach. Some comments/suggestions remain:

General:

I can relate to the authors aim of providing a polite and respectful critique of the MACH. However, in my opinion, in some cases this aim has led to formulations within the paper which do almost give an ambiguous notion about the usefulness of the scale, when in fact the usefulness of the scale could have heavily been called into question. I realize that a direct statement of this critique risks being interpreted as disrespectful – however, I think that the benefit, that readers/researchers get a clear-cut impression on the properties of the scale outweighs this risk. However, this is just a matter of opinion and there is no need for the authors to address this point (in fact, the revised version contained clearer formulations of the central properties of the scale.)

Specific:

l 190+: Maybe I do not understand this correctly, but the argument in terms of the negative loading does not seem convincing to me. If it is just a problem with respect to the computation of the sumscore, then one could resolve this issue by reversing the item (or by scoring it with a negative weight). A more convincing argument would deduce a contradiction between the quantity the factor is supposed to measure and the anticipated sign of the item that loads on this factor.

l 299+: It might be confusing to mention a sixth factor “However, the covariance between Factor 4 and Factor 6 was unidentified” within a five-factor model. Perhaps the authors could add a short statement to clarify this. (I guess, it refers to the elimination of the former fifth factor which was related to a Heywood case)

l 417+: I appreciate the added information on the treatment-control comparison. However, in order to avoid filtering the results in terms of statistical significance, I would recommend to provide a table which contains the results of the comparisons on all relevant items. That is, the table should also include the results of nonsignificant comparisons. Ideally, an estimate of an effect size measure (computed for significant as well as for nonsignificant comparisons) should also be added to enhance the evaluation of the treatment.

7. PLOS authors have the option to publish the peer review history of their article (what does this mean?). If published, this will include your full peer review and any attached files.

Reviewer #1: No

---

## [Author Response · Author response to Decision Letter 1]

9 Sep 2019

Response to Reviewers

Reviewer #1: The authors have addressed most of the points which were raised. They also provided convincing arguments for their approach. Some comments/suggestions remain:

General:

I can relate to the authors aim of providing a polite and respectful critique of the MACH. However, in my opinion, in some cases this aim has led to formulations within the paper which do almost give an ambiguous notion about the usefulness of the scale, when in fact the usefulness of the scale could have heavily been called into question. I realize that a direct statement of this critique risks being interpreted as disrespectful – however, I think that the benefit, that readers/researchers get a clear-cut impression on the properties of the scale outweighs this risk. However, this is just a matter of opinion and there is no need for the authors to address this point (in fact, the revised version contained clearer formulations of the central properties of the scale.)

Authors’ response: We thank the reviewer for understanding our plight. We “walked a tightrope” a bit but are glad that the reviewer is pleased with our revisions that do indicate our point of view of the scale.

Specific:

l 190+: Maybe I do not understand this correctly, but the argument in terms of the negative loading does not seem convincing to me. If it is just a problem with respect to the computation of the sumscore, then one could resolve this issue by reversing the item (or by scoring it with a negative weight). A more convincing argument would deduce a contradiction between the quantity the factor is supposed to measure and the anticipated sign of the item that loads on this factor.

Authors’ response: Ah…now we understand. We regret not being clearer in the original version of our paper when we simply listed "reverse coded" next to some example items. We actually recoded items that were reverse scored before submitting them to EFA and the CFA. We have added the following to lines 148-149: "All items were corrected for reverse scoring before the EFA was conducted.” Additionally, on lines 205-206 we added the following: “All items were corrected for reverse scoring before the CFA was conducted.” So, after reverse coding the reverse scored items we found that some of the items load negatively on some factors that also had positively loading items. This is not good in that items designed to measure facets of a construct should be positively correlated with that facet as well as with other items measuring the construct. Some of the items in the Mach IV are not positively correlated with each other. We thank the reviewer for pointing out how we may have just skipped by this without a proper explanation and we hope that our corrections to the paper and our explanation make things a bit clearer now.

l 299+: It might be confusing to mention a sixth factor “However, the covariance between Factor 4 and Factor 6 was unidentified” within a five-factor model. Perhaps the authors could add a short statement to clarify this. (I guess, it refers to the elimination of the former fifth factor which was related to a Heywood case)

Authors' response: This is a very good point. We have added the following to lines 277-278 in the hopes of clarifying our nomenclature: "In the following CFA we refer to the items of the Mach IV as well as the factors that emerged from the EFA by our original numbering system."

l 417+: I appreciate the added information on the treatment-control comparison. However, in order to avoid filtering the results in terms of statistical significance, I would recommend to provide a table which contains the results of the comparisons on all relevant items. That is, the table should also include the results of nonsignificant comparisons. Ideally, an estimate of an effect size measure (computed for significant as well as for nonsignificant comparisons) should also be added to enhance the evaluation of the treatment.

Authors' response: We have added the table that is suggested. In addition to the information requrested we have added a column in the table to indicate if an item was discarded or retained from the measurement invariance tests and a column indicating if the item was a revised item or an original item. To match this new Table 6 we have rewritten the verbiage on lines 415-425 as follows: "Additionally, in an item-by-item comparison of scores in the control and treatment groups, three items had unequal variances by virtue of Levene’s test. All were revised items. Item 6 about vicious streaks (F = 4.234, p < .05), item 11 about Barnum and suckers (F = 4.221, p < .05), and item 18 about men working hard (F = 18.130, p < .001) each had a larger spread of scores in the revised version than in the original version of the items. Item variances for other items in the control and treatment groups were not different. There were also significant differences in the means for those three items. Item 6 (t = -4.234, df = 472.66, p < .001, Cohen's d = .381) and item 18 (t = -6.425, df = 466.15, p < .001, d = .580) had higher means in the revised versions of the items and item 11 (t = 7.992, df = 476.49, p < .001, d = .727) had a higher mean in the original item. Item means for other items were not different. See Table 6 for the results for all items." Our new table begins on line 426. Thank you very much for your insightful suggestions and keen eye.

---

## [Editor Report · Decision Letter 2]

24 Sep 2019

Measurement invariance tests of revisions to archaically worded items in the Mach IV scale

PONE-D-19-15176R2

Dear Dr. Miller,

We are pleased to inform you that your manuscript has been judged scientifically suitable for publication and will be formally accepted for publication once it complies with all outstanding technical requirements.

With kind regards,

Angel Blanch, Ph.D.

Academic Editor

PLOS ONE
---

## [Editor Report · Acceptance letter]

2 Oct 2019

PONE-D-19-15176R2 

Measurement invariance tests of revisions to archaically worded items in the Mach IV scale 

Dear Dr. Miller:

I am pleased to inform you that your manuscript has been deemed suitable for publication in PLOS ONE. Congratulations! Your manuscript is now with our production department. 

With kind regards,

on behalf of

Dr. Angel Blanch 

Academic Editor

PLOS ONE